# RNA-Binding Protein MEX3A Interacting with DVL3 Stabilizes Wnt/β-Catenin Signaling in Endometrial Carcinoma

**DOI:** 10.3390/ijms24010592

**Published:** 2022-12-29

**Authors:** Pusheng Yang, Panpan Zhang, Shu Zhang

**Affiliations:** Department of Gynecology and Obstetrics, Shanghai Key Laboratory of Gynecology Oncology, Renji Hospital, Shanghai Jiao Tong University School of Medicine, Shanghai 200127, China

**Keywords:** RBP, MEX3A, endometrial carcinoma, DVL3, Wnt/β-catenin

## Abstract

Disease recurrence and metastasis lead to poor prognosis in patients with advanced endometrial carcinoma (EC). RNA-binding proteins (RBPs) are closely associated with tumor initiation and metastasis, but the function and molecular mechanisms of RBPs in EC are unclear. RBPs were screened and identified using the TCGA, GEO, and RBPTD databases. The effect of MEX3A on EC was verified by in vitro and in vivo experiments. Gene set enrichment analysis (GSEA), immunofluorescence (IF), and co-immunoprecipitation (Co-IP) were used to identify potential molecular mechanisms of action. We identified 148 differentially expressed RBPs in EC. MEX3A was upregulated and related to poor prognosis in patients with EC. In vitro and vivo experiments demonstrated that MEX3A promoted the growth, migration, and invasion capacities of EC cells. Mechanistically, DVL3, a positive regulator of the Wnt/β-catenin pathway, also increased the proliferation and metastasis of EC cells. MEX3A enhanced EMT and played a pro-carcinogenic role by interacting with DVL3 to stabilize β-catenin and upregulated the expression of its downstream target genes. MEX3A is upregulated in EC and promotes tumor progression by activating EMT and regulating the Wnt/β-catenin pathway via DVL3. MEX3A may therefore be a novel therapeutic target for EC**.**

## 1. Introduction

Due to declining fertility and rising obesity rates, the incidence of endometrial carcinoma (EC) has increased at a rate of around 1% per year. It is estimated that there will be approximately 65,950 people diagnosed with EC worldwide in 2022 [1,2]. Although patients with early stage EC have a better prognosis, minimal progress has been made in improving the survival of patients with this common tumor [3]. According to the American Cancer Society (ACS), the survival rate of patients with EC is 83.18% [4], while the 5-year survival rate of patients with advanced EC and muscular infiltration, lymphatic metastasis, and jumping distant metastases is less than 17% [5]. It was predicted that approximately 12,550 Americans would have died of EC by 2022 [6]. Since recurrence, metastasis, and high aggressiveness pose serious threats to patients with advanced endometrial cancer, it is critical to identify effective diagnostic targets and elucidate the molecular mechanisms underlying the proliferation and metastasis of EC.

Recently, several studies have indicated that RNA-binding proteins (RBPs) participate in cancer progression by regulating gene expression through post-transcriptional regulatory networks, including alternative splicing, translation, polyadenylation, the degradation of mRNAs, and processing of non-coding RNAs and microRNAs [7]. To date, approximately 2000 RBPs associated with cancer prognosis have been identified [8]. For example, Kaori Iino et al. demonstrated that the one RBP PSF increases the proliferation and metastasis of prostate and breast cancers through post-transcriptional regulation [9,10]. In colon cancer, PUM1 binds to the 3′-UTR of mRNA targets then promotes cell proliferation and migration [11]. In malignant peripheral nerve sheath tumors, the RBP HuR/ELAVL1 is highly expressed and drives tumor metastasis by interacting with multiple cancer-related transcripts [12]. Abnormal RBP functions therefore affect tumor progression, but the function of RBPs in EC has not been studied in depth.

Our bioinformatics analysis suggested that the Mex-3 RNA binding family member A (MEX3A), which is an RBP, may have an irreplaceable function in the progression of EC. MEX3A was originally identified to be a translation-regulating protein that maintains the germline cells’ totipotency [13]. It contains a RING domain and two K homology (KH) domains that can bind with RNA and participate in the regulation of RNA in various cancers. In lung adenocarcinoma, MEX3A is highly upregulated and promotes metastasis by interacting with LAMA2 [14]. Yang et al. found that MEX3A contributes to the proliferation and migration of glioma cells by targeting CCL2 [15]. In ovarian cancer, MEX3A has been found to promote malignant progression by guiding intron retention in TIMELESS [16,17]. Y. Wang et al. demonstrated that MEX3A regulated Wnt/β-catenin signaling in breast cancer [18]. Disheveled segment polarity protein 3 (DVL3) is an important multifunctional phosphoprotein of the Wnt/β-catenin pathway [19]. In our bioinformatic analysis, we found DVL3 was upregulated in EC. However, the regulatory mechanisms of MEX3A and DVL3 in EC have not yet been elucidated.

In our study, we comprehensively estimated the function of RBPs in EC and clarified the effects of MEX3A on the growth and metastasis of EC cells. We discovered that MEX3A was upregulated in EC and related to poor prognoses in patients. In vitro *and* in vivo experiments revealed that MEX3A significantly promotes the proliferation, migration, and invasion of EC cells. Additionally, we identified that DVL3 was also a potential tumor-promoting gene in EC. Mechanistically, MEX3A enhanced EMT and played a pro-carcinogenic role by interacting with DVL3 to stabilize β-catenin and upregulated the expression of its downstream target genes. Therefore, MEX3A may be a potential diagnostic and therapeutic target for treating patients with EC. Our study provides a molecular basis for early diagnosis and targeted treatment of EC patients.

## 2. Results

### 2.1. MEX3A, as a Representative RBP, Was Upregulated and Related to a Poor Prognosis in EC

We obtained 1970 RBPs with significant prognostic value from the RBPTD database and performed a differential analysis in EC. Then, a total of 148 RBPs were detected as differentially expressed RBPs (DE RBPs) in EC, of which 82 were upregulated and 66 were downregulated (Figure 1A). Detailed information on the DE RBPs is displayed in the volcano plot in Appendix A. Next, Kyoto Encyclopedia of Genes and Genomes (KEGG) and Gene Ontology (GO) analyses were conducted to assess the potential biological functions and mechanisms of the DE RBPs. We found that the DE RBPs were highly involved in the spliceosome, RNA degradation, RNA transport, mRNA surveillance, and other RNA-related pathways (Appendix A). Moreover, the GO analysis indicated many RNA-related cellular components and molecular functions, including ribonucleoprotein granules, spliceosomal complexes, P-body mRNA 3’-UTR binding, and ribonuclease activity (Appendix A). We also built a protein–protein interaction (PPI) network with 133 connections and 1126 edges of the DE RBPs (Appendix A). The key co-expressed module was selected by Molecular Complex Detection (MCODE) (Appendix A). Pathway and enrichment process analyses showed that the biological functions of the hub genes were primarily related to the processes of the spliceosome, RNA degradation, and RNA splicing (Appendix A). Based on these findings, we suspected that the abnormal expression of DE RBPs closely related to the molecular biological progression of EC.

To further explore the mechanisms of the 148 DE RBPs in EC, we carried out a comprehensive analysis based on four databases (DNA mutation, methylation, and protein and mRNA expression) of EC in the cBioPortal for Cancer Genomics website. The 148 DE RBPs were intersection analyzed with the top 200 genes derived from the DNA mutation, methylation, protein, and mRNA expression databases, respectively. Then, we obtained the only RBP, MEX3A, through the intersection analysis (Figure 1B). To investigate the role of MEX3A in EC, we first detected the expression pattern of MEX3A in different normal and tumor tissues from TCGA. MEX3A was preferentially expressed in the female reproductive tissues including ovary, cervix, and endometrium tissue in comparison to other 25 normal tissues. (Appendix A, *p* < 0.05). We further found that higher levels of MEX3A were expressed in various kinds of tumors (e.g., ovarian cancer, lung cancer, breast cancer, etc.) in comparison to normal tissues, and MEX3A in EC was significantly upregulated (Figure 1C, *p* < 0.05). In addition, the mRNA levels of MEX3A were identified as higher in the EC samples than in the normal endometrium based on three public databases (Figure 1D–F, *p* < 0.05). To further validate our results, we discovered that the protein expression of MEX3A in EC was also higher than in normal tissues (Appendix A, *p* < 0.05). Immunohistochemistry (IHC) staining also demonstrated that MEX3A was overexpressed in EC tissues (Figure 1G, *p* < 0.01). We also assessed the connection between the expression of MEX3A and patient outcomes. Kaplan–Meier (K–M) analysis indicated that the low level of MEX3A was positively associated with overall survival (OS) (Figure 1H, *p* < 0.05) and relapse-free survival (RFS) (Appendix A, *p* < 0.05) in patients with EC. These data indicate that MEX3A is crucial to the progression of EC.

### 2.2. MEX3A Promoted the Proliferation and Metastasis of EC Cells In Vitro

We measured the mRNA and protein levels of MEX3A in five EC cell lines (Ishikawa, HEC-1A, HEC-1B, ECC-1, and KLE), and the results showed that Ishikawa and HEC-1A exhibited a relatively high MEX3A expression level, while ECC-1 and HEC-1B had comparatively low MEX3A expression (Figure 2a,b, *p*  <  0.05). So, we selected Ishikawa and HEC-1A to knock down the MEX3A via short hairpin RNAs (shRNAs) transfection. After verifying the efficacy of silencing MEX3A in Ishikawa and HEC-1A (Figure 2c,d, *p*  <  0.05), cell proliferation ability was measured. The CCK-8 and colony-formation experiments demonstrated that the reduction of MEX3A diminished EC cells growth (Figure 2e,f, *p* < 0.05). Additionally, we investigated the role of MEX3A in EC cell metastasis. The transwell and wound-healing assays revealed that the knockdown of MEX3A reduced EC cells’ ability to migrate and invade. (Figure 2g,h, *p* < 0.05). Meanwhile, we overexpressed MEX3A in ECC-1 and HEC-1B cells, and the efficacy was assessed by qRT-PCR and Western blot (Appendix A, *p* < 0.05). As expected, overexpression of MEX3A promoted the proliferation, migration, and invasion of EC cells (Appendix A, *p* < 0.05). These findings illustrate that MEX3A enhances EC cell growth and metastasis in vitro.

### 2.3. MEX3A Increased the Oncogenesis and Progression of EC Cells In Vivo

To evaluate the effect of MEX3A in EC progression in vivo, we built a subcutaneous xenograft model by injecting Ishikawa-NC (NC) or Ishikawa-shMEX3A (shMEX3A) cells into nude mice. After 4 weeks of feeding, the mice in the NC group developed larger tumors than those in the shMEX3A group in terms of size, volume, and weight (Figure 3a–c, *p* < 0.05). Furthermore, the expression of Ki-67 declined in the sh-MEX3A group compared with that in the NC group, demonstrating that the proliferative ability of EC cells in vivo was reduced when MEX3A was inhibited. Meanwhile, the IHC of subcutaneous xenograft tumors also demonstrated that the MEX3A knockdown group considerably enhanced the expression of E-cadherin and decreased the expression of N-cadherin (Figure 3d–f, *p* < 0.05). A peritoneal metastasis model was then constructed to verify the impacts of MEX3A on EC metastasis in vivo. As expected, fewer metastatic nodules were observed in the MEX3A-knockdown group (Figure 3f–h, *p* < 0.05). Furthermore, in comparison to those in the NC group, the mice in the sh-MEX3A group were less likely to develop liver and lung metastases (Figure 3i–k, *p* < 0.05). Overall, these findings indicate that MEX3A promotes EC proliferation and metastasis in vivo.

### 2.4. MEX3A Activated the EMT and Wnt/β-Catenin Signaling Pathway in EC Cells

GSEA analysis showed that the EMT and Wnt/β-catenin signaling pathways were significantly enriched in the high-MEX3A-expression group (Figure 4a, *p* < 0.05). We therefore speculated that MEX3A participates in the EMT process via the Wnt/β-catenin signaling pathway to promote EC development. To validate our hypothesis, we evaluated the relationship between MEX3A and the EMT-related genes, E-cadherin, N-cadherin, vimentin, Snail, and Slug in TCGA-UCEC database. We found that the EMT-related genes were significantly related to MEX3A (Appendix A, *p* < 0.05). Next, qRT-PCR (Appendix A, *p* < 0.05) and Western blotting (Figure 4b, *p* < 0.05) were performed to analyze the impacts of MEX3A expression on the EMT-related genes. The results indicated that the knockdown of MEX3A led to an increase in the expression of E-cadherin and a decrease in the level of N-cadherin, vimentin, Snail, and Slug compared to the control in the Ishikawa and HEC-1A cells. As expected, MEX3A overexpression in the ECC-1 and HEC-1B cells resulted in opposite effects (Figure 4c and Appendix A, *p* < 0.05). These findings imply that MEX3A modulates the EMT pathway in EC.

We further evaluated the correlation between MEX3A and β-catenin, as well as its downstream targets c-MYC, CD44, and cyclin D1. The results showed a significant correlation between them (Appendix A, *p* < 0.05). The results of qRT-PCR and Western blot demonstrated that the mRNA and protein levels of β-catenin, c-MYC, CD44, and cyclin D1 were reduced following the knockdown of MEX3A in the Ishikawa and HEC-1A cells (Figure 4d and Appendix A, *p* < 0.05). The levels of these genes were notably upregulated by the overexpression of MEX3A in the ECC-1 and HEC-1B cells (Figure 4e and Appendix A, *p* < 0.05). To find out the role of MEX3A in the Wnt/β-catenin signaling pathway, we determined the subcellular localization of β-catenin. The results showed that the depletion of MEX3A led to a decline in the level of β-catenin in both the cytoplasm and nucleus, while the overexpression of MEX3A led to an increase in the cytoplasmic and nuclear β-catenin levels (Figure 4f). We therefore speculate that MEX3A may stabilize β-catenin expression against degradation and promote its entry into the nucleus, then enhance the transcription of c-MYC, CD44, and cyclin D1. These results support the hypothesis that MEX3A-mediated EMT in EC cells is primarily regulated by the Wnt/-catenin signaling pathway.

### 2.5. MEX3A Enhanced the Wnt/β-Catenin Signaling Pathway via DVL3

According to the median expression levels of MEX3A, we identified differentially expressed genes (DEGs) between the high and low groups (Appendix A, *p* < 0.05.), and then conducted functional enrichment analysis. KEGG and GO analysis demonstrated that the DEGs were greatly enriched in protein-containing complex disassembly, peroxisomes, and histone demethylase activity (Appendix A, *p* < 0.05). Next, we detected a relationship between MEX3A and 223 proteins in the Revers Phase Protein microArray (RPPA) database. Including DVL3, a total of 37 genes were identified significantly related to MEX3A. As Figure 5a showed, DVL3 had the strongest correlation with MEX3A (R = 0.59, *p* < 0.05). Since DVL3 is a critical component of the Wnt/β-catenin signaling pathway [20], we speculated that MEX3A may affect the Wnt/β-catenin signaling pathway via DVL3. To further examine the underlying mechanisms between the MEX3A and DVL3, we conducted in vitro experiments in EC cells. The Western blotting analysis implied that the protein level of DVL3 was greatly reduced after the reduction of MEX3A in Ishikawa and HEC-1A cells. The level of DVL3 was also increased when MEX3A was overexpressed in the ECC-1 and HEC-1B cells (Figure 5b). Additionally, the immunofluorescence assay demonstrated that DVL3 dots were merged with MEX3A points, illustrating the co-localization of DVL3 and MEX3A (Figure 5c). The co-IP experiments further demonstrated a direct interaction between MEX3A and DVL3 (Figure 5d,e). These findings illustrated that DVL3 is actively participated in the regulation of Wnt/β-catenin signaling pathway by MEX3A.

### 2.6. DVL3 Promoted the Progression of EC Cells In Vitro

Previous studies have found that DVL3, which is a multifunctional phosphoprotein of the Wnt/β-catenin signaling pathway, promotes the development of various malignancies by protecting β-catenin from degradation [21,22,23]. However, the biological function of DVL3 in EC is unclear. The analysis based on TCGA showed that EC tissues had higher levels of DVL3 expression compared to normal tissues (Figure 6a, *p* < 0.05). The IHC staining of the Ren Ji cohort further showed higher expression of DVL3 in the EC tissues (Figure 6b). The K–M analysis indicated that the expression level of DVL3 contributed to the prognosis of EC patients (Figure 6c, *p* < 0.05). Next, we constructed DVL3-knockdown and -overexpression cell lines to investigate the impacts of DVL3 on EC progression (Figure 6d and Appendix A). Then, CCK-8 and colony-formation assays showed that the proliferative capacity of Ishikawa and HEC-1A cells was impaired after the knockdown of DVL3 (Figure 6e,f, *p* < 0.05). The overexpression of DVL3, however, led to significant promotion of the growth of ECC-1 and HEC-1B cells (Appendix A, *p* < 0.05). We also assessed the metastatic properties of DVL3 in the EC cells. As expected, the migration and invasion ability of EC cells were significantly inhibited following the knockdown of DVL3 and enhanced following the overexpression of DVL3 (Figure 6e,f and Appendix A, *p* < 0.05). These findings imply that DVL3 has an influence on promoting EC progression.

According to the above results, we further investigated whether DVL3 is essential to the effect of MEX3A on EC cell growth and metastasis. We first determined the transfection efficacy of the specified cells via Western blotting (Figure 7a and Appendix A). As shown in Figure 7b,c, the overexpression of DVL3 partially reversed the decline in cell proliferation caused by the knockdown of MEX3A. However, reducing the level of DVL3 partially inhibited the pro-proliferative capacity of the cells due to MEX3A overexpression (Appendix A). In addition, DVL3 partially increased the migration and invasion capacities of the cells, which decreased when MEX3A was knocked down (Figure 7d,e). Meanwhile, when we knocked down DVL3 in ECC-1 and HEC-1B cells overexpressing MEX3A, the pro-metastatic effect of MEX3A was partially inhibited. (Appendix A, *p*  <  0.05). These findings illustrate that MEX3A may regulate the Wnt/β-catenin signaling pathway by interacting with DVL3.

In summary, the above results show that DVL3 is critical for the oncogenic influence of MEX3A on EC progression. By stabilizing the expression of DVL3, MEX3A suppressed the degradation activity of the degradation complex, stabilized free β-catenin in the cytoplasm, and promoted its entry into the nucleus, thereby activating the transcription of the downstream target genes CD44, c-Myc, and cyclin D1, and promoting the EMT pathway and EC cell proliferation and metastasis.

## 3. Discussion

RBPs are essential components of the post-transcriptional process that control the metabolism and expression of RNA through their specialized RNA-binding domains [24,25]. Recurrence and metastasis are the leading causes of death in patients with advanced EC [26]. Accumulating evidence has suggested that the dysregulation of RBPs is a hallmark of various tumors and is directly associated with tumor metastasis and patient prognoses [27,28,29]. In this research, a comprehensive analysis of the function of RBPs in EC was carried out, and we identified MEX3A as an actively expressed RBP that is strongly associated with patient prognoses. Several studies reported that MEX3A promoted the cancer progression of lung cancer, colorectal cancer, and ovarian cancer through the regulation of the mRNA expression of its target genes [14,17,30]. However, the roles and mechanisms of MEX3A in EC progression remain unclear. Our in vitro and in vivo experiments revealed that MEX3A significantly promotes the proliferation and metastasis of EC cells, suggesting that MEX3A may act as a novel target for EC diagnosis and therapy.

Increasing evidence has suggested that tumor development and metastasis are intimately correlated with EMT [31]. EMT is a dynamic process that is a critical step in the acquisition of invasive capacity by tumor cells, mainly in the form of the loss of polarity and gain of motility, which result in the transformation of epithelial cells into mesenchymal cells [32]. Several studies have suggested that the dysfunction of RBPs may lead to abnormal protein expression or function in the EMT pathway and activation of signaling pathways associated with tumor invasion, including the AKT, ERK, and Wnt pathways [33,34,35]. It has been claimed that MEX3A affects cell proliferation and migration via the EMT pathway in pancreatic and cervical cancers [36,37]. In this research, the overexpression of MEX3A encouraged the migration and invasion of EC cells by upregulating EMT-related genes including N-cadherin, vimentin, Snail, and Slug. The level of cell-adhesion molecules, such as E-cadherin, was significantly reduced after MEX3A was overexpressed. Therefore, MEX3A promotes EC cell proliferation and metastasis via the EMT pathway. The Wnt/β-catenin pathway largely participates in tumor metastasis and EMT [38,39,40]. We noticed that the knockdown of MEX3A caused a decline in the expression of β-catenin and its downstream target genes CD44, c-Myc, and cyclin D1. The β-catenin expression was significantly increased in both the nucleus and cytoplasm after MEX3A had been overexpressed. Therefore, we consider that MEX3A activates downstream target genes by stabilizing the expression of β-catenin and promoting its entry into the nucleus. We further determined that MEX3A interacts directly with DVL3 in EC cells.

As a member of the disheveled segment polarity proteins that act as signal amplifiers in the Wnt pathway, DVL3 significantly promotes the development of malignant tumors. In non-small cell lung cancer, DVL3 activates p65 to promote NSCLC progression through a p38-dependent pathway [41]. In esophageal squamous cell carcinoma, the knockdown DVL3 accounts for the inhibition of the growth and promotion of apoptosis of tumor cells [22]. Additionally, DVL3 acts as a modulator of resistance to IGFIR inhibition in breast and prostate cancer cells [42]. However, the effect of DVL3 in EC remains to be explored. To the best of our knowledge, this is the first study to explore the function of DVL3 in EC development. We found that DVL3, which is obviously upregulated in EC, was associated with poor prognosis in patients with EC and enhanced the proliferation, migration, and invasion of EC cells. Functional tests revealed that the upregulation of DVL3 led to the partial restoration of the reduced proliferation and migratory capacities of EC cells caused by the knockdown of MEX3A. Overall, MEX3A acted by interacting with DVL3 and affecting its expression, thus amplifying the impact of the Wnt/β-catenin pathway and subsequently promoting EC progression by promoting the EMT process.

## 4. Materials and Methods

### 4.1. Bioinformatics Analysis

The RNA-sequencing dataset and clinical information were obtained from The Cancer Genome Atlas (TCGA) database. A total 1970 RBPs with significant prognostic value according to the “survival” R package were obtained from RBPTD database (http://rbptd.com/). The differentially expressed genes of EC were, respectively, obtained from the GEPIA database and the GES 17,025 database by the linear model and the empirical Bayes (eBayes) method of the “limma” R package with adjusted *p*-value (Benjamini and Hochberg false-discovery rate, FDR). The screening criteria are *p* < 0.05 and |Log_2_Fold change (FC)| > 1. The DE RBPs were identified by taking the intersection with above three databases. The raw EC GeneChip data (GSE17025 and GSE183185) were downloaded from the Gene Expression Omnibus (GEO) database and processed by Sangerbox 3.0 [43]. Then, we detected the expression of MEX3A in normal and tumor tissues, with statistical significance determined by the *t*-test (*p* < 0.05). DNA mutation, methylation, protein, and mRNA expression profiles were acquired from cBioPortal. Differential analysis was performed using “DESeq2” R package and samples were divided into a “low” or “high” group based on the median expression levels of MEX3A in the TCGA-UCEC dataset (Appendix A). GO and KEGG enrichment analyses were performed by the “clusterProfiler” (version 3.14.3) R package based on the results of differential analysis. The inclusion criteria comprised |log_2_ FC| ≥1.0 and FDR < 0.05. The PPI network was extracted from the STRING 11.5 online database and visualized using Cytoscape 3.9.0 software. The hub genes and key modules were selected by the default parameter of “Cytohubba” and “MCODE” in Cytoscape software, respectively.

### 4.2. Immunohistochemistry (IHC) Assay

This study was approved by the Research Ethics Committee of Ren Ji Hospital, School of Medicine, Shanghai Jiaotong University (RA-2022-366). Forty paraffin-embedded specimens (twenty cases of EC and twenty cases of normal endometrium) were selected for the study. Image J was used to quantitatively examine the IHC results. The optical density (OD) was directly proportional to protein expression. The integrated optical density (IOD) was obtained by calculating the sum of the OD values at each point and represented the total amount of the target protein. The IOD value was divided by the area of the target protein distribution to obtain the average optical density (AOD), which was applied to contrast the distinctions in the expression of the target proteins. The protein expression formula was calculated as follows: AOD = IOD/area.

### 4.3. Cell Culture

Human EC cell lines (Ishikawa, HEC-1A, ECC-1, HEC-1B, and KLE) were obtained from the Cell Bank of the Chinese Academy of Sciences (Shanghai, China) and were cultured at 37 °C with 5% CO_2_. The HEC-1A cells were maintained in McCoy’s 5A medium, and the other cell lines were grown in RPMI 1640 medium with 10% fetal bovine serum (FBS, BI, USA) and 1% penicillin/streptomycin (P/S). All of the cells tested negative for mycoplasma contamination.

### 4.4. RNA Interference, Lentiviral Infection, and Plasmid Transfection

Short hairpin RNAs (shRNAs), specific small interfering RNAs (siRNAs), and lentiviruses for MEX3A and DVL3 were designed and synthesized by Genomeditech (Shanghai, China). Ishikawa and HEC-1A cells were infected with the shRNA lentivirus, while the OE-MEX3A or OE-DVL3 lentiviruses were introduced into the ECC-1 and HEC-1B cells. All of the cells were infected with lentiviruses for sh-negative control (NC) expression (sh-NC) or with a vector lentivirus (Vector) as a negative control. The cells were maintained for 2 weeks in medium containing puromycin dihydrochloride (Beyotime, Shanghai, China). Stable knockdown and overexpression cells were then used for further analysis. All target sequence information is demonstrated in Appendix A.

### 4.5. Quantitative Real-Time PCR (qRT-PCR) and Western Blot Analysis

qRT-PCR and Western blot analyses were performed as previously described [17]. The primers sequences of specific genes are listed in Appendix A. The detailed information of antibodies is shown in Appendix A.

### 4.6. Cell Proliferation Assay

Cell proliferation was measured by the CCK-8 assay (Share-bio, Shanghai, China). Absorbance was tested at 450 nm. For the colony formation assay, 1000 single cells were plated and maintained for 10–14 days. A microscope was used to count the colonies.

### 4.7. Migration and Invasion Assays

Transwell plates’ upper chambers (Corning, 3422) were seeded with 3 × 10^4^ cells in 200 μL of FBS-free medium, and 600 μL of 10% FBS medium was injected into the lower chamber. Cells were cultured for 24–48 h before being fixed with 4% paraformaldehyde and stained with 0.1% crystal violet. The top chambers were covered with Matrigel (1:8; BD, USA) prior to the invasion experiment. For the wound-healing test, a scratch gap was created using a 1000-μL pipette when cells had grown to 90% confluence. The wound area was detected 24 h and 48 h after scratching.

### 4.8. In Vivo Tumor Xenograft and Metastasis Assays

All experimental procedures were authorized by the Ethics Committee of the School of Medicine of Shanghai Jiaotong University (protocol number: A2019114). For the subcutaneous xenograft model, nude mice were subcutaneously injected with 5 × 10^6^ (100 μL) Ishikawa-NC or Ishikawa-shMEX3A cells in the axillae. The tumor diameters and mice’s weights were measured every 5 days for approximately 4 weeks. To induce experimental peritoneal metastasis, the peritoneal cavities of the nude mice were injected with 3 × 10^6^ Ishikawa-NC or Ishikawa-shMEX3A cells (100 μL). After 35 days, the mice were sacrificed and examined for tumor growth and metastasis.

### 4.9. Co-Immunoprecipitation (Co-IP)

In brief, the treated cells were lysed in IP buffer and incubated with equilibrated anti-Flag magnetic beads, protein A + G magnetic beads at 4 °C overnight, or rabbit IgG magnetic beads for 2 h at room temperature (RT) (BeyoMag™, China). The protein samples were then resuspended in sodium dodecyl sulfate loading buffer and detected via immunoblotting after culturing with the anti-Flag-MEX3A and anti-DVL3 antibodies.

### 4.10. Immunofluorescence Assay (IF)

The cells were seeded and cultured on glass coverslips the day before fixation with 4% paraformaldehyde, and then they were permeabilized for 10 min with 0.1% Triton X-100 and blocked for 1 h with 5% bovine serum albumin. The indicated primary antibodies were then incubated overnight at 4 °C. The next day, the cells were cultivated with the fluorescent secondary antibody for 1 h at RT. The DAPI was used to counterstain nuclei for 5 min. A Leica microscope was used to capture digital images.

### 4.11. Statistical Analysis

Each experiment was conducted at least thrice and represented as the mean ± standard deviation (SD). The data were examined using GraphPad Prism 8.0 and IBM SPSS Statistics 22. The survival of the patients based on the expression of MEX3A was evaluated using a K–M analysis. Student’s *t*-tests were applied to evaluate the difference between two groups. * *p* < 0.05, ** *p* < 0.01 and *** *p* < 0.001 were regarded as statistically significant.

## 5. Conclusions

In conclusion, our findings suggest a negative correlation between the level of MEX3A and the prognosis of patients with EC. Both MEX3A and DVL3 promoted EC cell proliferation, migration, and invasion. Mechanistically, the overexpression of MEX3A stabilized the expression of β-catenin by upregulating DVL3 and enhancing the EMT process to promote EC progression. Our findings enrich the understanding of the role of RBPs and identify MEX3A as a potential therapeutic target for EC. However, further research needs to be conducted to discuss the specific mechanisms underlying the interaction between MEX3A and DVL3 in EC.

## Figures and Tables

**Figure 1 ijms-24-00592-f001:**
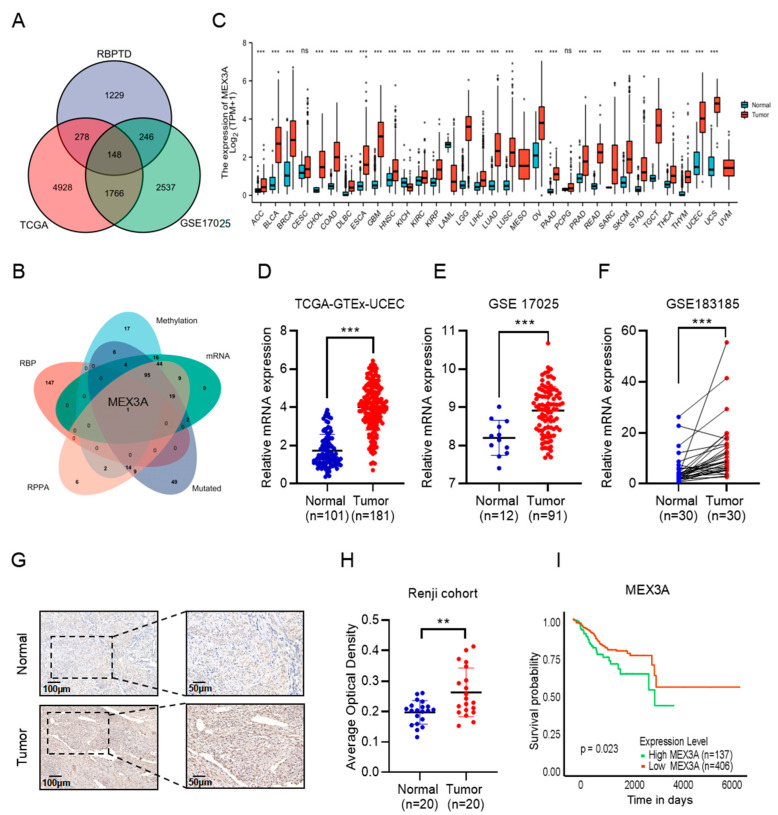
MEX3A is upregulated in EC and correlated with a poor prognosis in patients. (**A**) In total, 148 RBPs were identified as being differentially expressed in EC. (**B**) MEX3A was the only RBP identified by intersection analysis in EC. (**C**) Expression of MEX3A in 33 different tumors and corresponding normal tissues according to TCGA. (**D**–**F**) MEX3A expression in EC tissues and normal tissues in the TCGA-GTEx-UCEC, GSE17025, and GSE183185 datasets. (**G**) Immunohistochemistry stain of MEX3A expression in EC and normal tissues. Scale bars, 100 µm, 200 µm. (**H**) Immunohistochemical scoring of MEX3A expression based on images. AOD = IOD/area. (**I**) Kaplan–Meier analysis of the correlation between OS and the expression of MEX3A according to TCGA. ns *p*  >  0.05, ** *p * <  0.01, *** *p* < 0.001.

**Figure 2 ijms-24-00592-f002:**
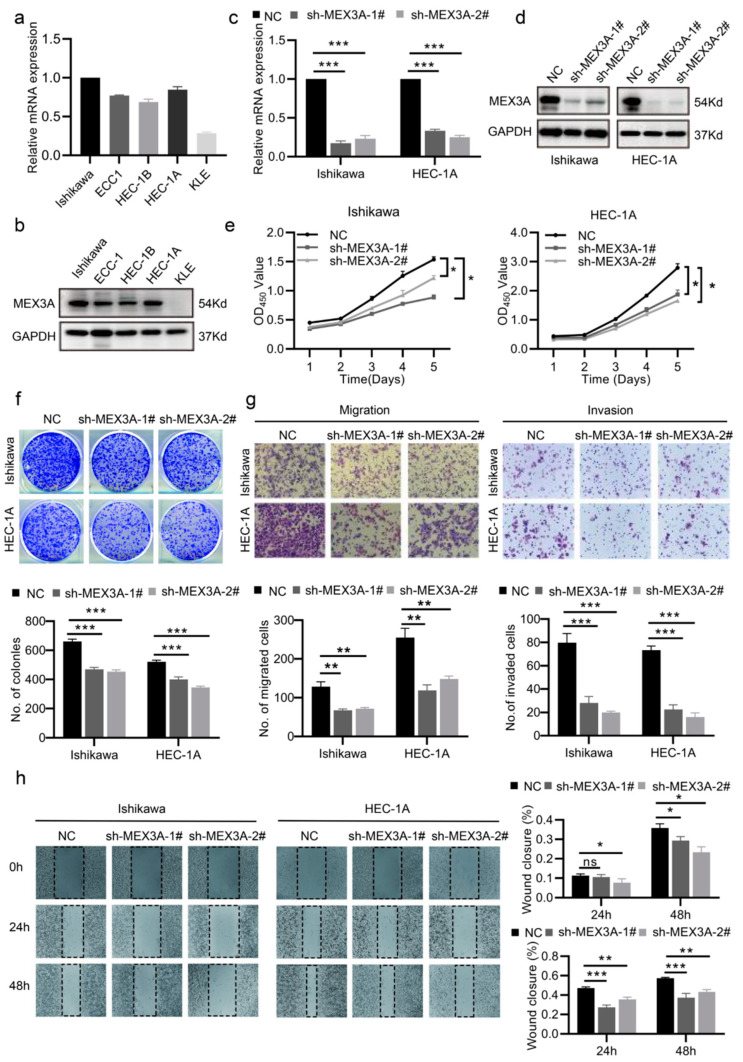
MEX3A promotes the proliferation and metastasis of EC cells in vitro. (**a**,**b**) mRNA and protein levels of MEX3A in five EC cell lines. (**c**,**d**) The efficacy of MEX3A knockdown in Ishikawa and HEC-1A cells measured through qRT-PCR and Western blotting. (**e**,**f**) CCK-8 and colony-formation assays (200×) showing that the MEX3A knockdown damaged the proliferation of Ishikawa and HEC-1A cells. (**g**,**h**) The migration and invasion abilities of Ishikawa and HEC-1A cells were impaired after the knockdown of MEX3A, as shown by the transwell and wound-healing assays (200×). * *p*  <  0.05, ** *p * <  0.01, *** *p* < 0.001.

**Figure 3 ijms-24-00592-f003:**
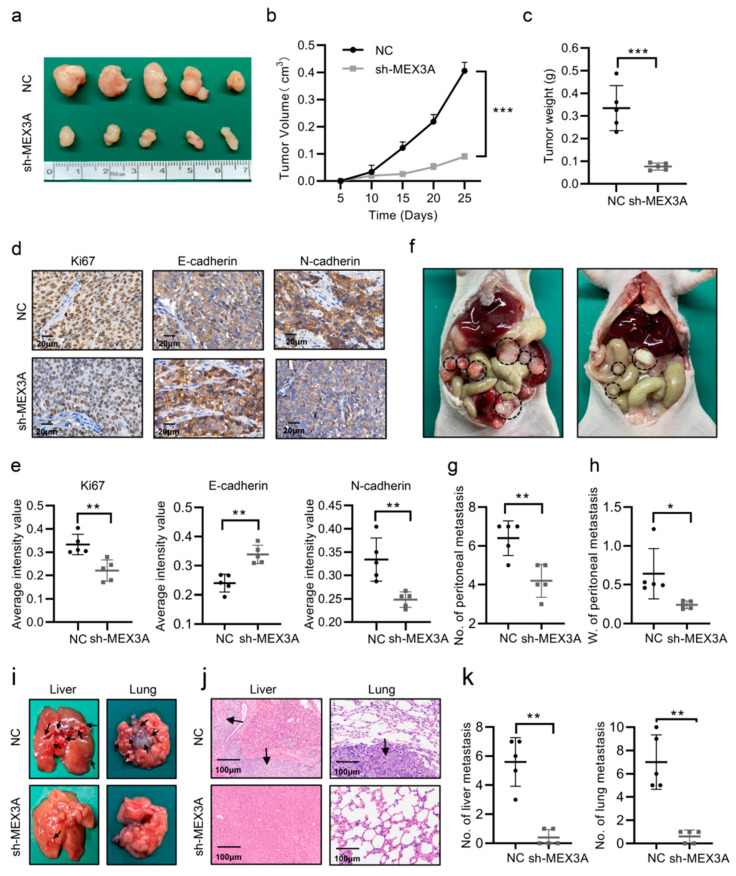
MEX3A increases the oncogenesis and progression of EC cells in vivo. (**a**) Growth of subcutaneous xenografts from control and MEX3A-knockdown cells. (**b**) The tumor volume of the subcutaneous xenografts was detected every 5 days. (**c**) After 25 days, the tumors were removed, weighed, and embedded in paraffin. (**d**,**e**) IHC staining of subcutaneous xenografts for Ki-67, E-cadherin, and N-cadherin. Scale bar, 20 μm. (**f**) Peritoneal metastasis model in nude mice injected with NC or sh-MEX3A cells. Black circles indicated the location of the metastatic nodules. (**g**,**h**) Numbers and weights of peritoneal metastases. (**i**) Representative images of metastatic sites in the liver and lung. Black arrows showed the location of the metastatic nodules. (**j**) HE staining of metastatic sites in the liver and lungs. Scale bar, 100 μm. (**k**) Number of metastatic nodules in the liver and lungs. * *p*  <  0.05, ** *p*  <  0.01, *** *p* < 0.001.

**Figure 4 ijms-24-00592-f004:**
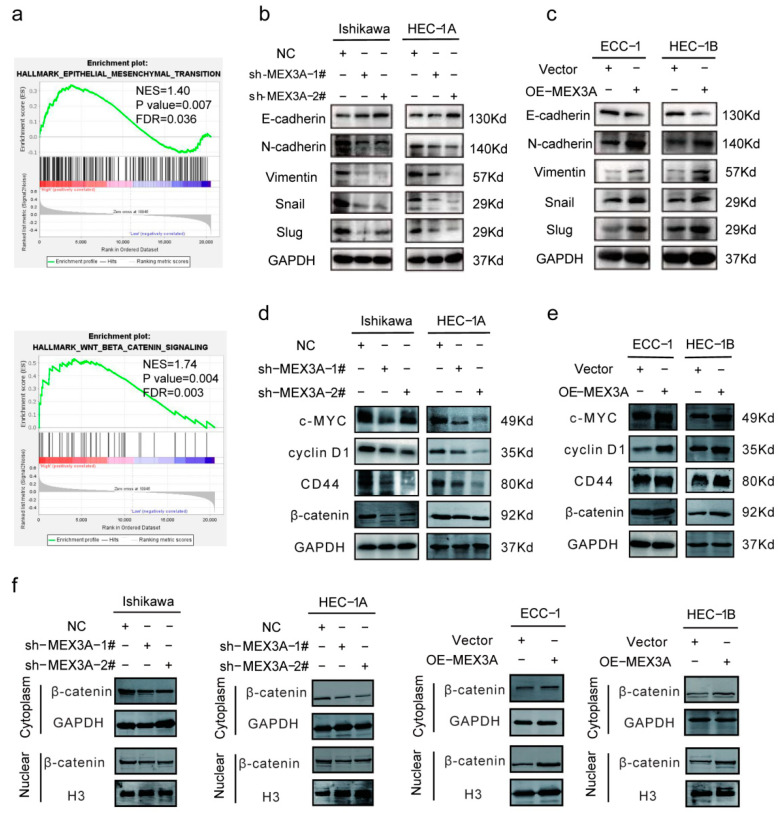
MEX3A activates the EMT and Wnt/β-catenin signaling pathway. (**a**) GSEA of the groups expressing MEX3A at high and low levels in TCGA. (**b**,**c**) The expression levels of EMT-related genes were investigated via Western blotting in both MEX3A-knockdown and -overexpression cell lines. (**d**,**e**) The protein levels of β-catenin and its downstream targets were explored via Western blotting in both MEX3A-knockdown and MEX3A-overexpression cells. (**f**) The role of MEX3A on nuclear and cytoplasmic β-catenin in MEX3A knockdown and overexpression cells was measured using Western blotting.

**Figure 5 ijms-24-00592-f005:**
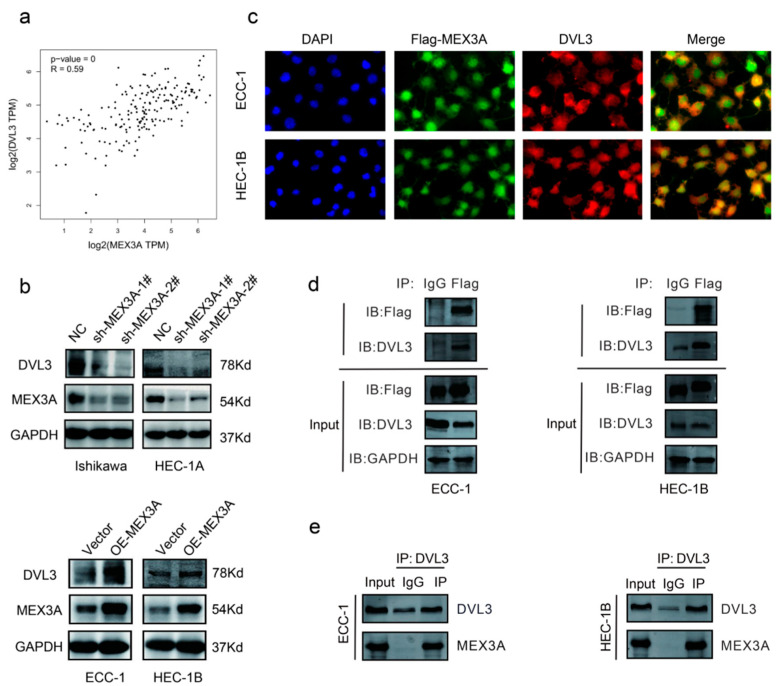
MEX3A enhances the Wnt/β-catenin signaling pathway via DVL3. (**a**) Relationship between MEX3A and DVL3 in TCGA (R = 0.59, *p* < 0.05). (**b**) The protein level of DVL3 after MEX3A knockdown or overexpression. (**c**) Immunofluorescence staining of Flag-MEX3A (green) and DVL3 (red) (400×). (**d**,**e**) Co-immunoprecipitation of MEX3A and DVL3.

**Figure 6 ijms-24-00592-f006:**
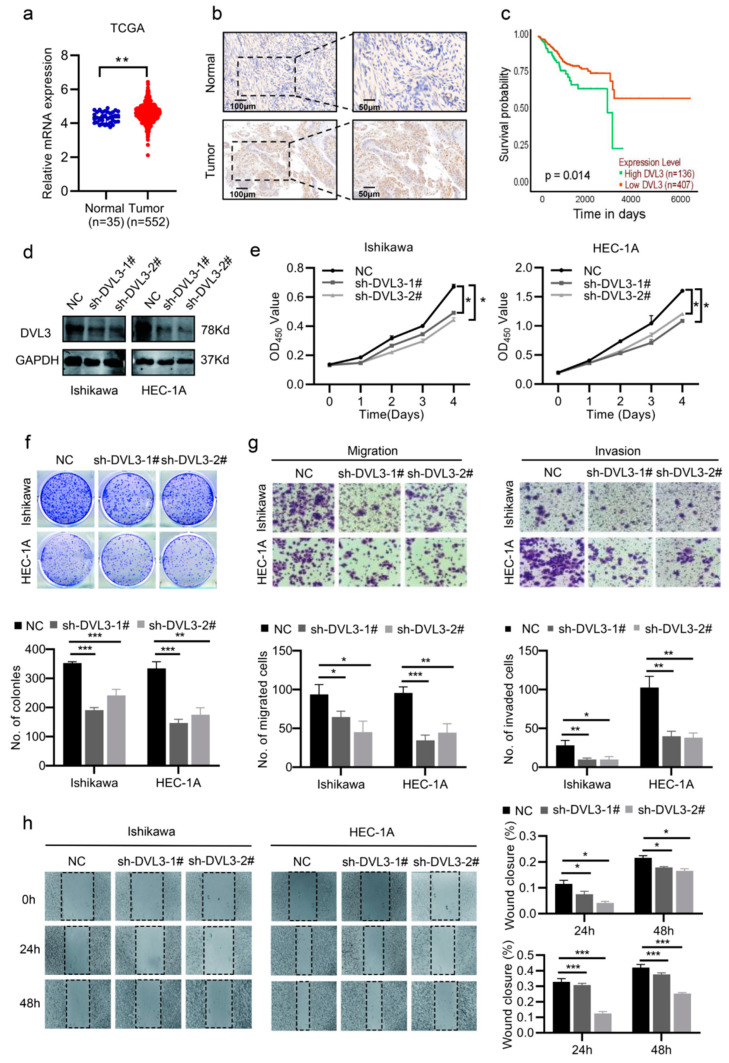
DVL3 promotes EC cell proliferation, migration, and invasion. (**a**) The mRNA levels of DVL3 were measured using data from TCGA-UCEC. (**b**) The IHC of DVL3 expression in EC and normal tissues in the Ren Ji cohort. Scale bar, 50 μm. (**c**) The survival of patients with EC was associated with the expression of DVL3 based on TCGA. (**d**) The efficacy of DVL3 knockdown was measured using Western blotting. (**e**,**f**) The CCK-8 and colony-formation assays (200×) showed that the proliferation of EC decreased after DVL3 knockdown. (**g**,**h**) The migration and invasion abilities of DVL3-knockdown EC cells were measured using transwell and wound-healing assays (200×). * *p*  <  0.05, ** *p*  <  0.01, *** *p* < 0.001.

**Figure 7 ijms-24-00592-f007:**
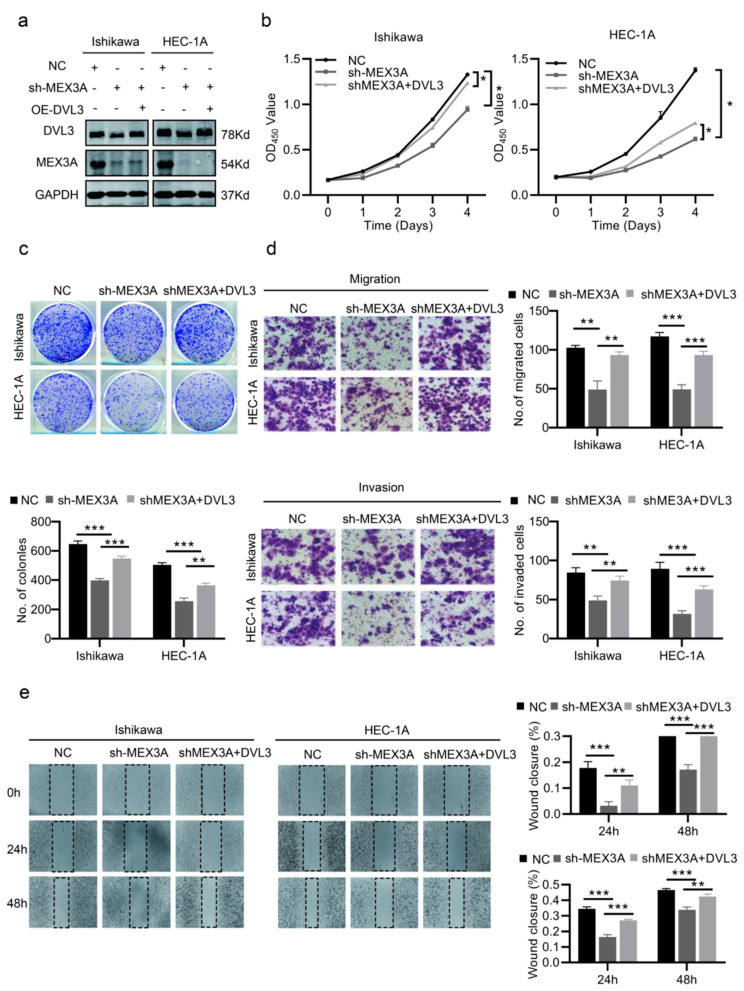
DVL3 is vital for the function of MEX3A in EC cell proliferation and migration. (**a**) The protein levels of MEX3A and DVL3 in Ishikawa and HEC-1A cells treated after MEX3A knockdown or DVL3 overexpression. (**b**,**c**) CCK-8 and colony-formation assays (200×) in different treatment groups (NC, sh-MEX3A, sh-MEX3A+ OE-DVL3). (**d**,**e**) Transwell and wound-healing assays performed with the indicated Ishikawa and HEC-1A cells (200×). * *p*  <  0.05, ** *p*  <  0.01, *** *p* < 0.001.

## Data Availability

Publicly available datasets were analyzed in this study. This data can be found here: https://www.cancer.gov/ (TCGA); https://www.ncbi.nlm.nih.gov/geo/ (GEO); http://rbptd.com/#/ (RBPTD); https://www.cbioportal.org/ (cBioPortal). All databases were accessed on 1 September 2021.

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
