# Peer review of "RNA-Binding Protein MEX3A Interacting with DVL3 Stabilizes Wnt/β-Catenin Signaling in Endometrial Carcinoma"

_ijms, 2022, doi:10.3390/ijms24010592_

Round 1
Reviewer 1 Report
Yang et al. reported that the role of MEX3A in stabilizing the Wnt/b-catenin signaling in endometrial carcinoma by interacting with DVL3. This is an interesting work. The experimental design of this work is scientific and rigorous, with sufficient arguments. Nevertheless, the current manuscript still needs be further improved before accepting for publish.
Major concerns:
1. Obviously, DVL3 is also an important gene in this study, but its description is less in the introduction section.
2. The Figures are not clearly, especially the Figure S1. please improve them.
3. In materials and methods section, the description of bioinformatic analysis lacks a lot of important information. In particular, many different databases and functional annotation and enrichment analyses were presented in this study. What is a reference genome used in this study? What software used to conducted the genome mapping ? How to do difference expression analysis, and so on. The authors should provide more detailed information.
4. Line 91-93. From this Figure, I can’t see any information on the MEX3A gene. Does the number 1 located at central represent the MEX3A? if so, I suggest you should mark it.
5. Line 170-171. I can’t see any evidence that supported this point. You should explain it Meanwhile, please provide the significant level.
Minior comments:
1. Line 9-10. please rewrite it.
2. Line 83. please provide full name for MCODE.
3. Line 103. please provide the full name for K-M analysis.
4. Line 120-121.please rewrite it.
5. Line 340.please provide database version
6. Line 343. please provide the ethic number
7. Line 356, please fix it (CO2).
8. Line 379,389,392, please fix it (104 and 106).
Author Response
Dear Reviewer:
We thank you very much for your positive and constructive comments concerning our manuscript entitled “RNA-binding protein MEX3A interacting with DVL3 stabilizes Wnt/β-catenin signaling in endometrial carcinoma” (ijms-2044138). Those comments are all valuable and very helpful for revising and improving our paper. We have addressed all of the points carefully and revised the manuscript according to your suggestions. Revised portions are marked in Red and responses in Blue.
Reviewer 1:
Yang et al. reported that the role of MEX3A in stabilizing the Wnt/b-catenin signaling in endometrial carcinoma by interacting with DVL3. This is an interesting work. The experimental design of this work is scientific and rigorous, with sufficient arguments. Nevertheless, the current manuscript still needs be further improved before accepting for publish.
Major concerns:
- Obviously, DVL3 is also an important gene in this study, but its description is less in the introduction section.
Response:
According to the reviewer’s thoughtful suggestions, we added the introduction of DVL3 in Lines 61-64 and Lines 69-74:
Line 61-64: Y. Wang, et al. demonstrated that MEX3A regulated Wnt/β-catenin signaling in breast cancer [18]. Disheveled Segment Polarity Protein 3 (DVL3) is an important multifunctional phosphoprotein of the Wnt/β-catenin pathway [19]. In our bioinformatic analysis, we found DVL3 was upregulated in EC. However, the regulatory mechanisms of MEX3A and DVL3 in EC have not yet been elucidated.
Line 69-74: Additionally, we identified that DVL3, a multifunctional phosphoprotein of the Wnt/β-catenin pathway, was also a potential tumor-promoting gene in EC. Mechanistically, MEX3A enhanced EMT and played a pro-carcinogenic role by interacting with DVL3 to stabilize β-catenin and up-regulated the expression of its downstream target genes.
- The Figures are not clearly, especially the Figure S1. please improve them.
Response:
According to the reviewer’s helpful suggestion, we have provided the images with higher resolution in our revised manuscript (New Figure 1, Figure S1, and Figure S4).
- In materials and methods section, the description of bioinformatic analysis lacks a lot of important information. In particular, many different databases and functional annotation and enrichment analyses were presented in this study. What is a reference genome used in this study? What software used to conducted the genome mapping? How to do difference expression analysis, and so on. The authors should provide more detailed information.
Response:
We thank reviewer’s valuable suggestion. We have given more description of bioinformatic analysis in Line 347-359:
The normalized RNA-sequencing dataset and clinical information were obtained from The Cancer Genome Atlas (TCGA) database. The DE RBPs were selected by the “limma” R package and visualized by the “ggplot2” R package based on the TCGA-UCEC database. We also downloaded EC GeneChip data (GSE17025 and GSE183185) from the Gene Expression Omnibus (GEO) database to detect the expression of MEX3A in normal and tumor tissues by the t-test (P<0.05). DNA mutation, methylation, protein and mRNA expression profiles were acquired from cBioPortal. The differential analysis was analyzed by the “DESeq2” R package based on the median expression levels of MEX3A in the TCGA-UCEC dataset. GO and KEGG enrichment analyses were performed by the “clusterProfiler” (version 3.14.3) R package based on the results of differential analysis. The exclusion criteria comprised |log2 FC| ≥1.0 and false discovery rate (FDR) < 0.05. The PPI network was extracted from the STRING 11.5 online database and visualized using Cytoscape 3.9.0 software. The hub genes and key modules were selected by the default parameter of “Cytohubba” and “MCODE” in Cytoscape software, respectively.
- Line 91-93. From this Figure, I can’t see any information on the MEX3A gene. Does the number 1 located at central represent the MEX3A? if so, I suggest you should mark it.
Response:
We appreciate the reviewer’s helpful suggestions. The number 1 located at central represents the MEX3A. We have added it in New Figure 1b.
- Line 170-171. I can’t see any evidence that supported this point. You should explain it Meanwhile, please provide the significant level.
Response:
As suggested by the reviewer, we have modified the result in Line 177-179 and re-uploaded a clear image of New Figure 4a:
GSEA analysis showed that the EMT and Wnt/β-catenin signaling pathways were significantly enriched in the high-MEX3A-expression group (Figure 4a, P < 0.05).
Minior comments:
- Line 9-10. please rewrite it.
Response:
According to the reviewer’s suggestion, we have rewritten it in Line 10-14:
Background: Disease recurrence and metastasis lead to poor prognosis in patients with advanced endometrial carcinoma (EC). RNA-binding proteins (RBPs) are closely associated with tumor initiation and metastasis, but the function and molecular mechanisms of RBPs in EC are unclear.
- Line 83. please provide full name for MCODE.
Response:
As the reviewer’s suggestion, we added the full name, Molecular Complex Detection, for MECOE in Line 90.
- Line 103. please provide the full name for K-M analysis.
Response:
According to the reviewer’s suggestion, we added the full name, Kaplan–Meier analysis, for K-M analysis in Line 108.
- Line 120-121.please rewrite it.
Response:
According to the reviewer’s valuable comment, we have rewritten it in Line 122-140:
We measured the mRNA and protein levels of MEX3A in five EC cell lines (Ishikawa, HEC-1A, HEC-1B, ECC-1, and KLE), and the results showed that Ishikawa and HEC-1A exhibited relatively high MEX3A expression level, while ECC-1 and HEC-1B had comparatively low MEX3A expression (Figure 2a–b, P < 0.05). So, we selected Ishikawa and HEC-1A to knock down the MEX3A by short hairpin RNAs (shRNAs) transfection. After verifying the efficacy of silencing MEX3A in Ishikawa and HEC-1A (Figure 2c–d, P < 0.05), cell proliferation ability was measured. The CCK-8 and colony formation experiments demonstrated that the reduction of MEX3A diminished EC cell growth (Figure 2e–f, P < 0.05). Additionally, we investigated the role of MEX3A in EC cells metastasis. The transwell and wound healing assays revealed that the knockdown of MEX3A reduced the ability of EC cells to migrate and invade. (Figure 2g–h, P < 0.05). Meanwhile, we overexpressed MEX3A in ECC-1 and HEC-1B cells, and the efficacy was assessed by qRT-PCR and Western blot (Figure S2A–B, P < 0.05). As expected, overexpression of MEX3A promoted the proliferation, migration, and invasion of EC cells (Figure S2C–F, P < 0.05). These findings illustrate that MEX3A enhances EC cell growth and metastasis in vitro.
- Line 340.please provide database version
Response:
According to the reviewer’s suggestion, we added the database version in Line 356: The PPI network was extracted from the STRING 11.5 online database and visualized using Cytoscape 3.9.0 software.
- Line 343. please provide the ethic number
Response:
The ethic number have added in Line 363:
This study was approved by the Research Ethics Committee of Ren Ji Hospital, School of Medicine, Shanghai Jiaotong University (RA-2022-366).
- Line 356, please fix it (CO2).
Response:
We have modified the spelling of CO2 in Line 375:
Human EC cell lines (Ishikawa, HEC-1A, ECC-1, HEC-1B, and KLE) were obtained from the Cell Bank of the Chinese Academy of Sciences (Shanghai, China) and were cultured at 37 °C with 5% CO2.
- Line 379,389,392, please fix it (104 and 106).
Response:
We thank the reviewer’s professional comments. We have fixed them in Line 400 and Line 410.
We appreciate for Editors/Reviewers’ warm work earnestly, the comments and suggestions are valuable and helpful for revising and improving our paper. We studied these comments carefully and hoped the correction will meet with approval. Once again, thank you very much for your comments and suggestions.
Yours, sincerely,
Shu Zhang

Reviewer 2 Report
The authors present the results of an interesting study on the effect of RNA-binding proteins on the progression of endometrial carcinoma, in particular MEX3A. The study also evaluated the regulation of MEX3A epithelial-mesenchymal transition and the Wnt/β-catenin pathway through interaction with DVL3.
The study includes various methods of analysis, both at the level of gene and protein expression, and at the level of an in vivo approach. The results of the study are important for the field of molecular oncology and may be of considerable interest to the readers of the journal.
Author Response
Dear Reviewer:
We thank you very much for your positive and constructive comments concerning our manuscript entitled “RNA-binding protein MEX3A interacting with DVL3 stabilizes Wnt/β-catenin signaling in endometrial carcinoma” (ijms-2044138). Those comments are all valuable and very helpful for revising and improving our paper. We have addressed all of the points carefully and revised the manuscript according to your suggestions. Revised portions are marked in Red and responses in Blue.
Reviewer 2:
The authors present the results of an interesting study on the effect of RNA-binding proteins on the progression of endometrial carcinoma, in particular MEX3A. The study also evaluated the regulation of MEX3A epithelial-mesenchymal transition and the Wnt/β-catenin pathway through interaction with DVL3.
The study includes various methods of analysis, both at the level of gene and protein expression, and at the level of an in vivo approach. The results of the study are important for the field of molecular oncology and may be of considerable interest to the readers of the journal.
Response:
We appreciate the reviewer’s positive and constructive comments on our manuscript. According to the reviewer’s suggestions, we have revised and put a lot of detail about the “Materials and Method”. Please see Line 347-359. We also polished the article again, and confirmation certificates from the editing service company (editage) were attached.
We appreciate for Editors/Reviewers’ warm work earnestly, the comments and suggestions are valuable and helpful for revising and improving our paper. We studied these comments carefully and hoped the correction will meet with approval. Once again, thank you very much for your comments and suggestions.
Yours, sincerely,
Shu Zhang

Reviewer 3 Report
This manuscript is not easy to judge. The topic is important and interesting. Results (figures) looks really convincing and experimental documentation is really good. However scientific paper has to present findings to wide community and must be easy to understand and follow. And this manuscript is not. English is difficult to understand. Often there is no logical connection between sentences. Description of several methods, especially bioinformatic analysis is not detailed enough (for instance I have no idea how authors divide the samples to MEX3A low and high expression, are they use median?). Authors use a lot of terms without clear explanation. Therefore I strongly recommend to use professional editing service and re-write the whole manuscript. Whenever it will be clearly written I will be happy to review again.
Author Response
Dear Reviewer:
We thank you very much for your positive and constructive comments concerning our manuscript entitled “RNA-binding protein MEX3A interacting with DVL3 stabilizes Wnt/β-catenin signaling in endometrial carcinoma” (ijms-2044138). Those comments are all valuable and very helpful for revising and improving our paper. We have addressed all of the points carefully and revised the manuscript according to your suggestions. Revised portions are marked in Red and responses in Blue.
Reviewer 3:
This manuscript is not easy to judge. The topic is important and interesting. Results (figures) looks really convincing and experimental documentation is really good. However scientific paper has to present findings to wide community and must be easy to understand and follow. And this manuscript is not. English is difficult to understand. Often there is no logical connection between sentences. Description of several methods, especially bioinformatic analysis is not detailed enough (for instance I have no idea how authors divide the samples to MEX3A low and high expression, are they use median?). Authors use a lot of terms without clear explanation. Therefore, I strongly recommend to use professional editing service and re-write the whole manuscript. Whenever it will be clearly written I will be happy to review again.
Response:
We thank the reviewer’s valuable comments. As reviewer’s suggestions, we modified the bioinformatic analysis in Line 342-354. We also have added a detail description of the terms, you can find these changes in Line 62, Line 90, and Line 108. We also carefully revised the whole manuscript for structure, consistency, clarity and conciseness, please see these changes in the revised manuscript, especially in Line11-14 (Abstract), Line 61-65 and Line 70-74 (Introduction), Line122-140, Line 217-229 and Line 277-280 (Result), Line 295-300 (Discussion), and Line347-359 (Materials and Methods).
Meanwhile, we polished our manuscript again, and confirmation certificates from the editing service company (editage) were attached.
We appreciate for Editors/Reviewers’ warm work earnestly, the comments and suggestions are valuable and helpful for revising and improving our paper. We studied these comments carefully and hoped the correction will meet with approval. Once again, thank you very much for your comments and suggestions.
Yours, sincerely,
Shu Zhang

Round 2
Reviewer 1 Report
No comments.
Author Response
Dear Reviewer:
We appreciate your review and comments earnestly,and sincerely hope our revised manuscript will meet with approval. Once again, thank you very much for your comments and suggestions.
Yours, sincerely,
Shu Zhang
Reviewer 3 Report
Manuscript was significantly improved and is much easier to understand. Experiments are well documented and explained. Importantly, Authors added part about bioinformatic analysis. And this part rise some concerns. In the lines 347-348 Authors state: ‘The normalized RNA-sequencing dataset and clinical information were obtained from The Cancer Genome Atlas (TCGA) database’. What do they mean by ‘normalized RNA-sequencing dataset’. Programs used for differential expression analysis usually use raw counts as input and normalize data themselves. For instance FPKM/TPM normalization should not be used as DE software input. To find DE RBPs they use limma package, which use log2CPM transformation, fit linear model to log2CPM, and calculate weights for each gene based on smoothed curve fitted to the sqrt(residual standard deviation) by average expression. Therefore its extremely important to provide correct input and Authors have to clarify this. They also have to provide more detailed description of analysis. How they construct expression matrix, namely how norm factors were calculated how genes with low expression were filtered out. Finally do they use eBayes or robustified eBayes procedure?
Lines 349-351: ‘We also downloaded EC GeneChip data (GSE17025 and GSE183185) from the Gene Expression Omnibus (GEO) database to detect the expression of MEX3A in normal and tumor tissues by the t-test (P<0.05)’ – It is unclear whether raw or processed data were downloaded. If raw, how they were processed, if processed – cite the paper(s) that describes how data were analyzed. Were the data from both datasets analyzed in the same/consistent way? If not, Authors have to reanalize them. I also assume that ‘to detect the expression of MEX3A in normal and tumor tissues by the t-test (P<0.05)’ means sth like: ‘to detect differences in the the expression of MEX3A in normal and tumor tissues, with statistical significance determined by the t-test (P<0.05)’. Lines 352-353: ‘The differential analysis was analyzed by the “DESeq2” R package based on the median expression levels of MEX3A in the TCGA-UCEC dataset’. Probably Authors mean ‘The differential analysis was performed using “DESeq2” R package and samples were divided to ‘low’ or ‘high’ group based on the median expression levels of MEX3A in the TCGA-UCEC dataset’. I have the same question as mentioned above – do Authors used raw counts as an input? My second concern is why Authors prefer using method based on linear model fitting to determine DE RBPs and method based on binomial distribution to determine genes differentially expressed in samples with ‘low’ vs ‘high’ MEX3A expression. Please clarify, why you use distinct methods for technically the same analysis or repeat both analyses using either limma or DESeq2. Authors should also provide supplementary tables with the lists of TCGA datasets used in DE analysis (for both DE RBPs, and MEX3A ‘high’ vs ‘low’). Lines 355-356: ‘The exclusion criteria comprised |log2 FC| ≥1.0 and false discovery rate (FDR) < 0.05’, probably Authors mean ‘inclusion’, not ‘exclusion’ criteria.
Minor comments:
Lines 80-81
Probably you mean that 148 RBP were detected as differentially expressed. Please correct sentence for the clarity. It will be also interesting to know what was the initial number of screened RBP and from which paper/analysis RBP list was obtained.
Lines 82-88
Probably you mean that ontology enrichment was conducted using KEGG and GO databases. It is expected that when you run GO/KEGG enrichment using list of RBP as an input you will obtain processes related to RNA metabolism/life cycle as an output. This analysis doesn’t bring anything important to the manuscript, consider removing it.
Lines 93-94
‘These results indicated that RBPs act as an important role in RNA metabolism and protein translation’ - one doesn’t need to run any analysis to state that RBPs play important role in these processes.
Line 95-98
It is unclear how you select MEX3A, what are the results of mentioned analysis for other RBPs?
Lines 100-101
I don’t think that editing service did great job if they kept sentences like this unedited: ‘The expression of MEX3A was upregulated in the female reproductive tissues than in the others’. It will be better to write: ‘The expression of MEX3A was upregulated in the female reproductive tissues in comparison to the other tissues’. How many tissues were compared? What was the significance level? Please show this comparison in supplementary figure.
Lines 107-110
‘The Kaplan–Meier (K–M) analysis indicated that the level of MEX3A was positively associated with overall survival (OS) (Figure 1h, P < 0.05) and relapse-free survival (RFS) (Figure S1I, P < 0.05) in patients with EC. These data indicate that MEX3A is crucial to the progression of EC. ‘ - this sentence is not true. Authors should clearly state that the low level of MEX3A was positively associated with overall survival and relapse-free survival.
Lines 112-113
‘(b) MEX3A was selected as a special RBP in EC’ – explain why
Author Response
Dear Reviewer:
We thank you very much for your positive and constructive comments concerning our manuscript entitled “RNA-binding protein MEX3A interacting with DVL3 stabilizes Wnt/β-catenin signaling in endometrial carcinoma” (ijms-2044138). Those comments are all valuable and very helpful for revising and improving our paper. We have addressed all of the points carefully and revised the manuscript according to your suggestions. The “Round 1” revised portions are marked in Red, the “Round 2” revised portions are marked in Purple and responses are in Blue.
Manuscript was significantly improved and is much easier to understand. Experiments are well documented and explained. Importantly, Authors added part about bioinformatic analysis. And this part rise some concerns. In the lines 347-348 Authors state: ‘The normalized RNA-sequencing dataset and clinical information were obtained from The Cancer Genome Atlas (TCGA) database’. What do they mean by ‘normalized RNA-sequencing dataset’. Programs used for differential expression analysis usually use raw counts as input and normalize data themselves. For instance FPKM/TPM normalization should not be used as DE software input. To find DE RBPs they use limma package, which use log2CPM transformation, fit linear model to log2CPM, and calculate weights for each gene based on smoothed curve fitted to the sqrt(residual standard deviation) by average expression. Therefore its extremely important to provide correct input and Authors have to clarify this. They also have to provide more detailed description of analysis. How they construct expression matrix, namely how norm factors were calculated how genes with low expression were filtered out. Finally do they use eBayes or robustified eBayes procedure?
Response:
We thank the reviewer’s valuable comments. We have revised our manuscript and given more description of selecting DE RBPs in Line 360-368:
The RNA-sequencing dataset and clinical information were obtained from The Cancer Genome Atlas (TCGA) database. A total 1970 RBPs with significant prognostic value by “survival” R package were obtained from RBPTD database (http://rbptd.com/). The differentially expressed gene of EC were respectively obtained from GEPIA database and GES 17025 database by the linear model and the empirical Bayes (eBayes) method of the “limma” R package with adjusted p-value (Benjamini and Hochberg false discovery rate, FDR). The screening criteria are P<0.05 and |Log2Fold change (FC)| > 1. The DE RBPs were identified by taking the intersection with above three databases.
Lines 349-351: ‘We also downloaded EC GeneChip data (GSE17025 and GSE183185) from the Gene Expression Omnibus (GEO) database to detect the expression of MEX3A in normal and tumor tissues by the t-test (P<0.05)’ – It is unclear whether raw or processed data were downloaded. If raw, how they were processed, if processed – cite the paper(s) that describes how data were analyzed. Were the data from both datasets analyzed in the same/consistent way? If not, Authors have to reanalize them. I also assume that ‘to detect the expression of MEX3A in normal and tumor tissues by the t-test (P<0.05)’ means sth like: ‘to detect differences in the the expression of MEX3A in normal and tumor tissues, with statistical significance determined by the t-test (P<0.05)’.
Response:
We thank the reviewer’s professional comments. As reviewer’s suggestions, we have carefully modified the part of “Bioinformatics analysis” in Line 368-371:
The raw EC GeneChip data (GSE17025 and GSE183185) were downloaded from the Gene Expression Omnibus (GEO) database and processed them by Sangerbox 3.0 [45]. Then we detect the expression of MEX3A in normal and tumor tissues, with statistical significance determined by the t-test (P<0.05).
Lines 352-353: ‘The differential analysis was analyzed by the “DESeq2” R package based on the median expression levels of MEX3A in the TCGA-UCEC dataset’. Probably Authors mean ‘The differential analysis was performed using “DESeq2” R package and samples were divided to ‘low’ or ‘high’ group based on the median expression levels of MEX3A in the TCGA-UCEC dataset’. I have the same question as mentioned above – do Authors used raw counts as an input? My second concern is why Authors prefer using method based on linear model fitting to determine DE RBPs and method based on binomial distribution to determine genes differentially expressed in samples with ‘low’ vs ‘high’ MEX3A expression. Please clarify, why you use distinct methods for technically the same analysis or repeat both analyses using either limma or DESeq2. Authors should also provide supplementary tables with the lists of TCGA datasets used in DE analysis (for both DE RBPs, and MEX3A ‘high’ vs ‘low’). Lines 355-356: ‘The exclusion criteria comprised |log2 FC| ≥1.0 and false discovery rate (FDR) < 0.05’, probably Authors mean ‘inclusion’, not ‘exclusion’ criteria.
Response:
According to the professional comments of the reviewers, we carefully examined the R programs and revised this part in Line 362-372:
A total 1970 RBPs with significant prognostic value by “survival” R package were obtained from RBPTD database (http://rbptd.com/). The differentially expressed gene of EC were respectively obtained from GEPIA database and GES 17025 database by the linear model and the empirical Bayes (eBayes) method of the “limma” R package with adjusted p-value (Benjamini and Hochberg false discovery rate, FDR). The screening criteria are P<0.05 and |Log2FC| > 1. The DE RBPs were identified by taking the intersection with above three databases. The raw EC GeneChip data (GSE17025 and GSE183185) were downloaded from the Gene Expres-sion Omnibus (GEO) database and processed them by Sangerbox 3.0 [45]. Then we detected the expression of MEX3A in normal and tumor tissues, with statistical significance determined by the t-test (P<0.05). DNA mutation, methylation, and protein and mRNA expression profiles were acquired from cBioPortal. The differential analysis was performed using “DESeq2” R packageand samples were divided to ‘low’ or ‘high’ group based on the median expression levels of MEX3A in the TCGA-UCEC dataset (Table S4) [46].
Reference:
[45] Shen, Weitao, Ziguang Song, Xiao Zhong, Mei Huang, Danting Shen, Pingping Gao, Xiaoqian Qian, Mengmeng Wang, Xiubin He, Tonglian Wang, Shuang Li, and Xiang Song. "Sangerbox: A Comprehensive, Interaction-Friendly Clinical Bioinformatics Analysis Platform." iMeta 1, no. 3 (2022): e36.
[46] Vasaikar, S. V., P. Straub, J. Wang, and B. Zhang. "Linkedomics: Analyzing Multi-Omics Data within and across 32 Cancer Types." Nucleic Acids Res 46, no. D1 (2018): D956-D63.
Minor comments:
Lines 80-81
Probably you mean that 148 RBP were detected as differentially expressed. Please correct sentence for the clarity. It will be also interesting to know what was the initial number of screened RBP and from which paper/analysis RBP list was obtained.
Response:
As reviewer’s suggestions, we have modified this part in Line 80-83 and Line361-364:
Line 80-83: We obtained 1,970 RBPs with significant prognostic value from the RBPTD database and performed a differential analysis in EC. Then a total of 148 RBPs were detected as differentially expressed RBPs (DE RBPs) in EC, of which 82 were upregulated and 66 were downregulated (Figure 1a).
Line361-364: A total 1970 RBPs with significant prognostic value by “survival” R package were obtained from RBPTD database (http://rbptd.com/).
Lines 82-88
Probably you mean that ontology enrichment was conducted using KEGG and GO databases. It is expected that when you run GO/KEGG enrichment using list of RBP as an input you will obtain processes related to RNA metabolism/life cycle as an output. This analysis doesn’t bring anything important to the manuscript, consider removing it.
Response:
We thank the reviewer’s valuable comments. In Line 80-97, we aimed to perform a comprehensive analysis of RBPs in EC and fond out potential RBP for further studies. First, we identified differentially expressed RBPs (DE RBPs) associated with prognosis in EC. We then focused on finding out which RNA-associated pathways are regulated in EC by these DE RBPs during EC progression. Therefore, we performed the KEGG and GO analyses. This part of the findings provided the basis for our subsequent study of RBPs in EC, and for the completeness of the article, we have put this part of the results in Supplementary Materials (Figure S1).
Lines 93-94
‘These results indicated that RBPs act as an important role in RNA metabolism and protein translation’ - one doesn’t need to run any analysis to state that RBPs play important role in these processes.
Response:
According to reviewer’s comments, we have modified the results in Line 94-97:
Based on these findings, we suspected that the abnormal expression of DE RBPs closely related to the molecular biological progression of EC.
Line 95-98
It is unclear how you select MEX3A, what are the results of mentioned analysis for other RBPs?
Response:
We thank the reviewer’s thoughtful comments. We have given more description of selecting MEX3A in Line 98-105:
To further explore the mechanisms of the 148 DE RBPs in EC, we carried out a comprehensive analysis based on four databases (DNA mutation, methylation, and protein and mRNA expression) of EC in the cBioPortal for Cancer Genomics website. The 148 DE RBPs were intersection-analyzed with the top 200 genes derived from the DNA mutation, methylation, protein, and mRNA expression databases, respectively. Then, we obtained the only RBP, MEX3A through the intersection analysis (Figure 1b).
Lines 100-101
I don’t think that editing service did great job if they kept sentences like this unedited: ‘The expression of MEX3A was upregulated in the female reproductive tissues than in the others’. It will be better to write: ‘The expression of MEX3A was upregulated in the female reproductive tissues in comparison to the other tissues’. How many tissues were compared? What was the significance level? Please show this comparison in supplementary figure.
Response:
According to reviewer’s helpful suggestions, we have rewritten it in Line 100-102, and added the image in Fiugre S1h:
MEX3A was preferentially expressed in the female reproductive tissues including ovary, cervix, and endometrium tissue in comparison to other 25 normal tissues. (Figure S1h, P<0.05).
Lines 107-110
‘The Kaplan–Meier (K–M) analysis indicated that the level of MEX3A was positively associated with overall survival (OS) (Figure 1h, P < 0.05) and relapse-free survival (RFS) (Figure S1I, P < 0.05) in patients with EC. These data indicate that MEX3A is crucial to the progression of EC. ‘ - this sentence is not true. Authors should clearly state that the low level of MEX3A was positively associated with overall survival and relapse-free survival.
Response:
We thank the reviewer’s valuable comments. We have modified it in Line 118-120:
The Kaplan–Meier (K–M) analysis indicated that the low level of MEX3A was positively associated with overall survival (OS) (Figure 1h, P < 0.05) and relapse-free survival (RFS) (Figure S1I, P < 0.05) in patients with EC.
Lines 112-113
‘(b) MEX3A was selected as a special RBP in EC’ – explain why
Response:
According to the reviewer’s thoughtful suggestions, we have revised the figure legend of Figure 1b in Line 125-126:
(b) MEX3A was the only RBP identified by intersection analysis in EC.
To screen the potential RBP that may be closely related to endometrial cancer progression, we performed intersection analysis on 148 RBPs as well as top 200 genes in the four datasets (DNA mutation, methylation, and protein and mRNA expression). The result showed that MEX3A was identified as the only RBP in EC (Figure 1b). Therefore, we chose MEX3A for the subsequent studies. According to reviewer’s suggestions, we have modified this part in Line 98-105:
To further explore the mechanisms of the 148 DE RBPs in EC, we carried out a comprehensive analysis based on four databases (DNA mutation, methylation, and protein and mRNA ex-pression) of EC in the cBioPortal for Cancer Genomics website. The 148 DE RBPs were intersection-analyzed with the top 200 genes derived from the DNA mu-tation, methylation, protein, and mRNA expression databases, respectively. Then, we obtained the only RBP, MEX3A through the intersection analysis (Figure 1b).
We appreciate for Editors/Reviewers’ warm work earnestly, the comments and suggestions are valuable and helpful for revising and improving our paper. We studied these comments carefully and hoped the correction will meet with approval. Once again, thank you very much for your comments and suggestions.
Yours, sincerely,
Shu Zhang
Round 3
Reviewer 3 Report
Thank you for addressing all my comments. Now I understand every step of the analysis. I am impressed by short response time and recommend to accept your manuscript.